# Selecting the Most Important Features for Predicting Mild Cognitive Impairment from Thai Verbal Fluency Assessments

**DOI:** 10.3390/s22155813

**Published:** 2022-08-03

**Authors:** Suppat Metarugcheep, Proadpran Punyabukkana, Dittaya Wanvarie, Solaphat Hemrungrojn, Chaipat Chunharas, Ploy N. Pratanwanich

**Affiliations:** 1Department of Computer Engineering, Faculty of Engineering, Chulalongkorn University, Bangkok 10330, Thailand; 6270302421@student.chula.ac.th; 2Department of Mathematics and Computer Science, Faculty of Science, Chulalongkorn University, Bangkok 10330, Thailand; dittaya.w@chula.ac.th (D.W.); naruemon.p@chula.ac.th (P.N.P.); 3Department of Psychiatry, Faculty of Medicine, Chulalongkorn University, Bangkok 10330, Thailand; solaphat@hotmail.com; 4Cognitive Fitness and Biopsychological Technology Research Unit, Chulalongkorn University, Bangkok 10330, Thailand; 5Cognitive Clinical & Computational Neuroscience Research Unit, Department of Internal Medicine, Faculty of Medicine, Chulalongkorn University, Bangkok 10330, Thailand; chaipat.c@chula.ac.th; 6Chula Neuroscience Center, King Chulalongkorn Memorial Hospital, Thai Red Cross Society, Bangkok 10330, Thailand; 7Chula Intelligent and Complex Systems Research Unit, Chulalongkorn University, Bangkok 10330, Thailand

**Keywords:** MoCA, mild cognitive impairment, phonemic verbal fluency, feature extraction, silence-based feature, similarity-based feature, phonemic clustering, switching, classification

## Abstract

Mild cognitive impairment (MCI) is an early stage of cognitive decline or memory loss, commonly found among the elderly. A phonemic verbal fluency (PVF) task is a standard cognitive test that participants are asked to produce words starting with given letters, such as “F” in English and “ก” /k/ in Thai. With state-of-the-art machine learning techniques, features extracted from the PVF data have been widely used to detect MCI. The PVF features, including acoustic features, semantic features, and word grouping, have been studied in many languages but not Thai. However, applying the same PVF feature extraction methods used in English to Thai yields unpleasant results due to different language characteristics. This study performs analytical feature extraction on Thai PVF data to classify MCI patients. In particular, we propose novel approaches to extract features based on phonemic clustering (ability to cluster words by phonemes) and switching (ability to shift between clusters) for the Thai PVF data. The comparison results of the three classifiers revealed that the support vector machine performed the best with an area under the receiver operating characteristic curve (AUC) of 0.733 (N = 100). Furthermore, our implemented guidelines extracted efficient features, which support the machine learning models regarding MCI detection on Thai PVF data.

## 1. Introduction

Thailand entered an aging society in 2001 when the aging population over 65 was around 7% of the country’s population. By 2050, its aging population is expected to reach 35.8%; i.e., ~20 million people [1]. In 2021, the share of population older than 65 years old in Thailand accounted for 12.4%. According to prevalence studies, mild cognitive impairment (MCI) was found in ~20% of the elderly [2,3,4]. This percentage is alarming to healthcare professionals because MCI causes a cognitive change in people over 65 years of age that can develop into Alzheimer’s disease (AD) or dementia [5]. Early detection of MCI is essential for the elderly to manage their lifestyle, which may alleviate the impairments in brain function [4]. However, a diagnosis of MCI can be time consuming and cost intensive due to the need for several clinical procedures. Using information and communication technology will facilitate clinicians to overcome these limitations.

The Montreal cognitive assessment (MoCA) is a prominent screening assessment tool to diagnose cognitive impairment [6,7,8,9,10]. MoCA is used to diagnose MCI by considering patients’ performance in various cognitive functions using tests. Inevitably, MoCA has some limitations. First, the original paper-and-pencil MoCA requires experts to conduct the assessment with the participants. Second, it cannot be used for sightless or motor disabilities. Third, the assessment result is stored manually only on paper, making it difficult to further analyze the results.

A possible solution to mitigate the above limitations is to consider the analysis of verbal fluency (VF). There are two categories of VF: semantic VF (SVF) and phonemic VF (PVF). Many scholars have shown the success of MCI detection using VF [11,12,13,14,15,16]. SVF can be obtained when patients are asked to say a word in certain categories (e.g., fruits, animals). Meanwhile, for PVF, MoCA prompts patients to say words beginning with specific letters, such as “F”, in 1 min. The score of a PVF test is calculated from the total number of correct answers. A decline in VF or a low score is evidence of frontal lobe dysfunction, which is related to the symptoms of MCI [17]. The number of generated words in Thai PVF substantially differs between MCI and a healthy control (HC) [18]. Several studies have suggested ways to extract features from PVF for MCI detection, which will be extended in the related work.

Although the abovementioned analytical process performs well in English, it cannot be applied to Thai. The main reason is that Thai has different grammatical rules and structures compared with English [19], which could pose numerous problems, such as (1) the problem of phonemic clustering in Thai, which requires subcategories to be rearranged; (2) the homophone problem because Thai has several sets of letters that produce the same sound, differing in its definitions (e.g., “กรรณ” /kan/, “กัน” /kan/); (3) the compound word problem due to prefixing, i.e., “การ” /kaan/ or “กระ” /krà/, to change the types or definitions of words (e.g., “การบ้าน” /kaanˑbâan/, “การเรียน” /kaanˑrian/, “กระโดด” /kràˑdòot/, and “กระรอก” /kràˑrɔ̂ɔk/); (4) the tonal characteristic that adds challenges to speech recognition (e.g., “ก่อน” /kɔ̀ɔn/, “ก้อน” /kɔ̂ɔn/); and (5) the consonant cluster problem for groups of two consonants, i.e., “กล” /kl/, “กร” /kr/, and “กว” /kw/, that make a distinct sound in pronunciation (e.g., “กล้าม” /klâam/, “กราบ” /kràap/, “กวาด” /kwàat/). Naturally, there must be novel methods to remedy these problems. Due to the linguistic characteristics, we have noticed this vulnerability. We plan to use our Thai language proficiency to address these issues.

In this study, we focused on detecting MCI using Thai PVF data from the digital MoCA [10], which has validity as assessed by examining Spearman’s rank order coefficients and the Cronbach alpha value [20]. To solve the language barriers, we planned to use our proficiency in Thai language to develop a novel phonemic clustering and switching algorithm. Furthermore, we proposed a novel method by combining various feature types with feature selection using the chi-square test. In this way, we achieved a promising result in detecting MCI using Thai PVF data and highlighted the feature’s importance for further research investigation.

## 2. Related Work

VF tasks are employed for assessing neuropsychology because of their conciseness and ease of use. Participants are asked to name as many words as possible in 1 min under a given condition. SVF has the condition of requiring participants to identify things, such as animals or fruits. Meanwhile, PVF has the condition of requiring participants to produce words beginning with specific letters, such as F or P. Several scholars have analyzed variants within VF tasks to observe the processes that influence cognitive impairment.

Troyer et al. [21] introduced two essential components in VF: clustering—the grouping of words within semantic or phonemic subcategories; and switching—the ability to transition between clusters. Ryan et al. [22] compared cognitive decline between experienced boxers and beginners and proposed a cluster using a similarity score of phonemes in VF. They showed that the number of fights was significantly related to shifting ability. Mueller et al. [23] investigated the correlation between PVF and SVF using data from the Wisconsin Registry for Alzheimer’s Prevention. They showed that persons with amnestic MCI poorly have lower scores than the control group. Clustering is related to the tendency for participants to produce words within the same category. Switching refers to participants’ conscious decision to shift from one category to another [24].

Word similarity is an effective strategy for detecting cognitive impairment. Levenshtein et al. [25] introduced the Levenshtein distance (LD) to evaluate word similarity by edit distance. LD is the number of operations (e.g., insertions, deletions, and substitutions) required for transforming one word into another. Orthographic similarity, calculated from comparing letters in words, is commonly used in psycholinguistics; it involves lexical access in word memory [24,25,26]. Semantic similarity is based on word meaning or definition; it affects letter fluency performance, such as the degradation of nonverbal conceptual information [27]. Lindsay et al. [28] proposed alternative similarity metrics (e.g., LD, weighted phonetic similarity, weighted position in words, and semantic distance between words, clustering, and switching) with a two-fold evaluating argument. They showed that weighted phonemic edit distance had the best result for assessment in PVF. Further, similarity-based features have been reported to help improve model accuracy by 29% for PVF [29].

Spontaneous speech is a sensitive parameter to identify cognitive impairment in VF. Hoffmann et al. [30] proposed four temporal parameters of spontaneous speech by Hungarian native speakers. Their examination included the hesitation ratio, articulation rate, speech tempo, and grammatical errors. They showed that the hesitation ratio is the best parameter for identifying AD. However, measuring these parameters can be time consuming. T’oth et al. [31] performed automatic feature extraction using automatic speech recognition (ASR) for laborious processes. Their method, which could be used as a screening tool for MCI, yielded an F1-score of 85.3. Using silence-based features (e.g., silence segment, filled pauses, and silence duration) with a machine learning technique has yielded an F1-score of 78.8% for detecting MCI [15]. Recently, Campbell et al. [32] proposed an algorithm based on analyzing the temporal patterns of silence in VF tasks using the “AcceXible” and “ADReSS” databases. Their results showed that the silence-based feature had the best accuracy in the VF tasks. Several studies within the same scope have indicated that silenced-based features are the biomarkers for detecting cognitive impairment [13].

In conclusion, the abovementioned features (silenced-based features, similarity-based features, and clustering) are related to cognitive decline in MCI. These features have different capabilities and implications in discrimination. We found the possibility to integrate them with state-of-the-art machine learning techniques in MoCA application for medical benefits and some improvement. However, some features may be unsuitable for Thai, which we investigate in this study.

## 3. Materials and Methods

In this section, we provide an overview of our experiment. Our experiment includes data collection, feature extraction, classification, feature selection, and results (Figure 1).

### 3.1. Data Collection

Participants were assessed via MoCA application for their cognition (Figure 2) [10]. Voice data were recorded in .m4a file format at 44.1 kHz, 32 bits, via an iPad’s microphone.

In this paper, we used data from a PVF task in which participants were asked to name as many words as possible in 1 min from a given letter, “ก” /k/. Participants were categorized into two groups by the MoCA score: the HC group, with an MoCA score of 25 or above, and the MCI group, with an MoCA score of less than 25. The participants’ demographics are presented in Table 1. All participants were Thai native speakers and provided consent before the assessment began.

### 3.2. Feature Extraction

Feature extraction is the process of extracting useful information from data, such as audio and transcribed files. Figure 1 represents the diagram of our extracting process. Table 2 shows the features we used and their description.

#### 3.2.1. Silenced-Based Features

After recording the participant’s voice, voice activity detection was used to detect the presence or absence of human speech for further calculation of voice features, such as the average silence between words and total silence. In this study, the silent and voice segments were measured using the Pydub Python package [33]. Further, background noise and irrelevant conversation were removed before processing. All the calculation methods for the silence-based features consisted of the basic mathematics described in Table 2.

#### 3.2.2. Similarity-Based Features

The similarity in the word list was computed based on its orthography or semantics by comparing the target word with the next word. The comparison was continued until the last member of the list, and then the average similarity was calculated from the summation divided by the list length. In this study, we computed semantic similarity using the PyThaiNLP Python package [34]. In addition, LD was computed according to the method reported in the original research article [25]. The orthography similarity has a slightly modified calculation method, which is explained in the Section 3.2.3.

#### 3.2.3. Orthographic Similarity in Thai

The orthographic similarity assigns a number between 0 and 1, indicating the similarity of words, where 1 means that words are similar, whereas 0 is dissimilar [26]. We employed the original method to calculate the similarity in Thai words, but the vowel in Thai can be written above or below the letter. Thus, the calculation procedure was slightly modified, as shown in Figure 3.

#### 3.2.4. Phonemic Clustering for Thai PVF

Phonemic clustering is the word production inside the phoneme [21,35]. Clustering depends on temporal lobe functions, such as word storage and working memory. Therefore, we decided to group words according to Thai characteristics [19]. We started by anticipating the possibilities that a word will generate in the letter fluency task “ก” /k/. After knowing all the possibilities, we decided to group words into four different categories, as represented in Table 3. The Thai language is a tonal language, and the way it is written and pronounced are different from others. Accordingly, the algorithm for classifying words into clusters needs to be redesigned, as explained in detail in Appendix A.

#### 3.2.5. Switching in Thai

Switching is the ability to transition between word clusters [21,35]. Switching depends on frontal lobe functions, such as the searching strategy, shifting, and cognitive flexibility. A switching-based feature is calculated by counting the number of transitions between the phonemic clusters (Figure 4).

For the workflow, the word at the first position is determined as the fourth cluster (C4), and the next word is determined as the second cluster (C2). After comparing adjacent words, add 1 to the switching score if the words are in different clusters. The process is repeated until the last word. Notably, Figure 4 has a switching score of 5 and a clustering score of 4.

### 3.3. Classification

Classification is the process of class prediction from given data, where the classes refer to the targets or labels. This work investigated two class labels: the MCI and HC groups, labeled 1 and 0, respectively. We employed extreme gradient boosting (XGBoost), support vector machine (SVM), and random forest (RF) as the classifiers. We also applied the 10-fold cross-validation technique to reduce data biases.

In this study, we used the scikit-learn Python library [36], which is an open-source and efficient tool for predictive data analysis.

### 3.4. Feature Selection

Feature selection was used for model simplification, training time reduction, and model accuracy increment [37]. In this paper, we selected features according to the chi-square value ($\chi^{2}$) via the Chi2 algorithm [38]. The $\chi^{2}$ test indicates a relationship between each feature and the class label, which is MCI. Typically, it can be assumed that the lower the $\chi^{2}$, the more correlated it is with the class label. The formula for calculating the $\chi^{2}$ value is
(1)$$\chi_{c}^{2}=\sum^{\;}\frac{{\left(O_{i}-E_{i}\right)}^{2}}{E_{i}}\;\;$$
where $O_{i}$ is the observed value of the feature, and $E_{i}$ is the expected value of class label, which is MCI.

### 3.5. Evaluation

Six standard measures were used to evaluate the model performance: Accuracy measures the percentage of correct prediction, as shown in (2).
(2)$$\mathrm{Accuracy}=\frac{\mathrm{TP}+\mathrm{TN}}{\mathrm{TP}+\mathrm{TN}+\mathrm{FP}+\mathrm{FN}}$$

Precision defines the percentage of MCI that the model correctly predicted (3), whereas recall is the ratio that requires a closer look at false positives (4).
(3)$$\mathrm{Precision}=\frac{\mathrm{TP}}{\mathrm{TP}+\mathrm{FP}}$$
(4)$$\mathrm{Recall}=\frac{\mathrm{TP}}{\mathrm{TP}+\mathrm{FN}}$$

For a simple comparison of these two values, the F1-score, the harmonic mean of precision and recall, is considered (5).
(5)$$F1\text{-}\mathrm{score}=\frac{2\times \mathrm{Precision}\times \mathrm{Recall}}{\mathrm{Precision}+\mathrm{Recall}}$$
where true positive (TP) is the actual MCI that the model predicted as MCI, false positive (FP) is the normal that the model predicted as MCI, true negative (TN) is the normal that the model predicted as normal, and the false negative (FN) is the actual MCI that the model predicted as normal.

The area under the receiver operating characteristic curve (AUC) is an effective method for summarizing the diagnostic accuracy across all possible decision thresholds [39]. Typically, AUC ranges from 0 to 1, an AUC of 0.5 implies random prediction, 0.7–0.8 is considered acceptable, 0.8–0.9 is considered excellent, and >0.9 is considered outstanding. This study emphasizes an AUC interpretation in light of research evidence suitable for disease classification [39,40].

## 4. Results

### 4.1. Classification Results

All features were trained and tested into three classifiers (XGBoost, SVM, and RF) with 10-fold cross-validation. Table 4, Table 5 and Table 6 show the classification results for each set of features. It can be observed that the best classifier is SVM, with an AUC of 0.733 with nine features, whereas the other statistical values are inconsistent. This result can be attributed to the numerous true negatives in the prediction process, as can be seen with the specificity of 0.883. The set of seven features, more consistent for practical use, provides an acceptable result at an AUC of 0.729. Meanwhile, the acceptable result for the SVM features is between 5 and 7. RF reveals the most accurate prediction, with an AUC of 0.683 with 11 features. Meanwhile, XGBoost provide the best result at 0.671 with 13 features. These results confirm our hypothesis that the Thai PVF can distinguish MCI patients and HC individuals.

### 4.2. Feature Importance

In this section, we computed the prediction value of each feature using Shapley additive explanations (SHAP), an algorithm for ranking the features that impact the classification results [41].

Figure 5 shows two excellent features for the RF classifier: the average silence between the words and the number of silence segments. Low values of the average silence between words affected the model from −0.16 to 0.00, whereas medium-to-high values affected the model from 0.00 to 0.01. In contrast, high values of the number of silence segments affected our model from −0.14 to 0.00, whereas low-to-medium values had an effect from 0.00 to 0.05.

Figure 6 illustrates two excellent features for XGBoost: the average silence between words and switching. High values of the average silence between words affected our model from 0 to 1, whereas low values affect from 2 to 0. In contrast, high values of switching affected our model from −1.2 to 0, whereas low values had an effect from 0 to 1.

Figure 7 shows that switching and the different silence between Q1 and Q2 had a good prediction power. A high switching value affected our model from −1.2 to 0, whereas low values had an effect from 0 to 1. Similarly, medium-to-high and low values of the different silence between Q1 and Q2 affected the model from −0.2 to 0.0 and 0 to 0.4, respectively.

In summary, the SHAP algorithm shows the impact on the model using the concept of game theory, which helps to interpret the feature’s value and understand the model decision. Figure 5, Figure 6 and Figure 7 represent feature ranking in each classifier; the average silence between words and switching is ranked at the top in every classifier. Moreover, these results are consistent with the chi-square test during the feature-selection process (see Figure 8). The chi-square test reveals the p-value based on the dependent hypothesis between the feature and class; it shows seven features with a low *p*-value to convey the idea. Accordingly, this stack of five-to-seven features reasonably yields the maximum accuracy in SVM.

## 5. Discussion

The goals of the present study were to use the data from the Thai PVF task for MCI detection and develop the guidelines for clustering in the feature extraction for Thai PVF. Using state-of-the-art machine learning techniques with optimal feature extraction produced acceptable results for MCI detection (Table 4, Table 5 and Table 6).

### 5.1. Feature Analysis

Our findings provide three pieces of evidence that are consistent with previous research. First, the prediction value of the silenced-based feature for MCI detection is high [30]. The average silence between words is ranked at the top of the SHAP values. Silence might be accounted for by impaired processes of lexical access and word-finding difficulties. MCI tends to have extended silence before saying the next word, whereas silence in the PVF task directly implicates the number of generated words. Figure 9 shows that MCI’s box and HC’s box of the average silence between words are almost symmetric. The median indicates that the data between HC and MCI are likely different. Second, the prediction value of switching is high, but clustering is not (Figure 5 and Figure 7). This finding agrees with the original research that switching is more essential than clustering for optimal performance on PVF, whereas switching and clustering were equally essential for SVF [21]. Switching involves the transition between clusters. Alternatively, switching may be related to the ability to initiate a search for a new strategy or subcategory. MCI seems to have a lower value of switching compared with HC. Figure 9 shows that the median of the MCI box is almost outside the HC box, suggesting that the two groups are different. Third, similarity-based features seem to have no prediction value. Similarity-based features were ranked almost last in terms of feature importance (Figure 5, Figure 6 and Figure 7). Semantic similarity, which involves producing a different vocabulary, reveals the best p-value in the chi-square test compared with other similarity features. Figure 9 shows that the MCI box is sparse. Furthermore, the median of the HC box is within the MCI box, indicating that this feature is inappropriate for MCI detection. These results correspond to those of a previous study that the semantic feature and LD had a worse silhouette coefficient than Troyer’s proposed method [28].

### 5.2. Classification Analysis

In this study, three classifiers were chosen based on their algorithm’s basis and advantages in a performance comparison. SVM is advantageous in high-dimensional data, and it can customize kernel functions to transform data into a required form. RF is based on several decision tree classifiers on various subsamples of a dataset and uses averaging to improve the predictive accuracy [36]. XGBoost is based on the gradient boosting algorithms, optimized and distributed to be highly efficient, flexible, and portable [42].

We found that SVM is the best classifier among the three. Furthermore, we obtained slightly better results when increasing the number of significant features in the classification process (Table 4, Table 5 and Table 6), which agrees with a previous study [15]. Additionally, we performed fine-tuning to choose the optimal parameters in each classifier. From the result, we suggest that each classifier should be used for a task that it is good in. Therefore, SVM is suitable for widespread use because it has the highest AUC, which is a threshold-free evaluation metric. Meanwhile, RF performs stably even when increasing the number of features; the AUC is between 0.617 and 0.683. XGBoost’s performance is similar to that of RF, with an AUC between 0.617 and 0.671. Furthermore, in terms of training data and fine-tuning, XGBoost is the fastest among the three classifiers.

### 5.3. Limitations and Future Work

Our proposed phonemic clustering and switching guidelines demonstrate the benefits of MCI detection for Thai native speakers. This proposal fills the gap between the differences in language characteristics. Our algorithms are also simplified and do not require high computing power, which is suitable for a mobile or small device. Accordingly, we believe this guideline will aid in the cost-effective automation of MCI detection.

However, this study has some limitations. First, our data were obtained from only one type of Thai PVF. Another Thai VF assessment (fruit categories, animal categories, and other letters, such as /S/ “ส”) has not been investigated yet. Next is the small and unbalanced dataset. Unfortunately, we collected data for this research during the coronavirus outbreak. Thus, there were insufficient participants to collect a large amount of data due to the lockdown policy. Finally, high-accuracy ASR for PVF is needed to handle a large amount of data. Several text-to-speech solutions perform well in typical situations; e.g., when speaking long sentences. However, when applied to an audio clip using PVF, unacceptable results were realized. Maybe the PVF does not have the context clues to help the computer speculate the next word. Further, PVF has so many short-speech styles that it is difficult to specify whether they are phonemes or tones. Besides, the Thai language has different word meanings using tones. For this reason, the more accurate the text-to-speech solution, the more extensive data we can handle.

For future research, we developed the digital MoCA to collect beneficial information during a test. We plan to use the data from other tasks (backward digit span, serial sevens, and memory test) obtained from the digital MoCA. We believe that selecting a significant feature from the various tasks will encourage the performance of MCI detection or other relevant diseases (dementia and AD). We also plan to use the Thai text-to-speech solution [10] that focuses on PVF in terms of being fully automated.

## 6. Conclusions

In this study, we focused on detecting MCI by using data from Thai PVF, which is essential due to the growth of the ageing population in Thailand. Our method gave an acceptable result of MCI detection by combining various feature types via chi-square feature selection with an AUC of 0.733. We examined the valuable feature of the machine learning model to distinguish between HC and MCI for Thai PVF. Moreover, we introduced the guideline for phonemic clustering and the initial approach for measuring the similarity between words for Thai PVF, which is proven to be consistent with previous research. We believe that our findings will be helpful for further practical implementation and development.

## Figures and Tables

**Figure 1 sensors-22-05813-f001:**
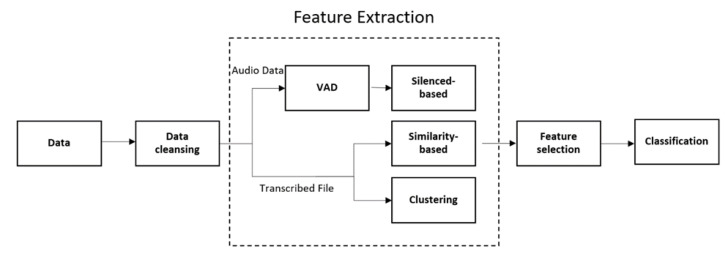
Our machine learning framework.

**Figure 2 sensors-22-05813-f002:**
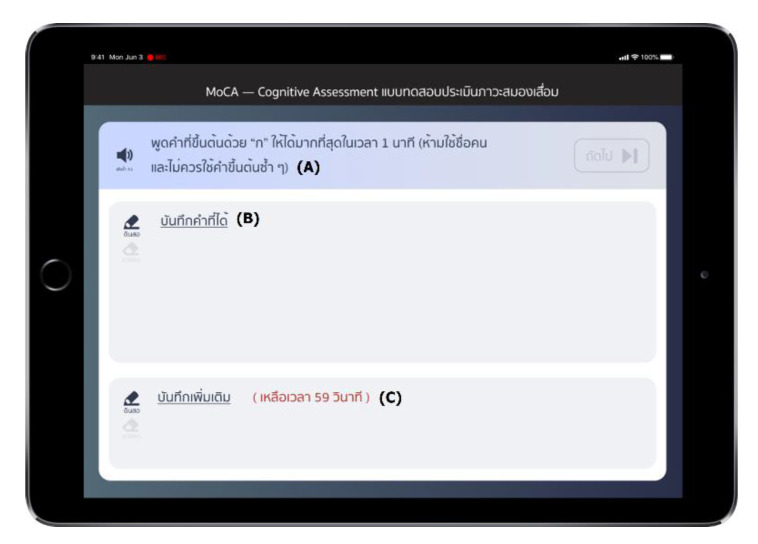
The PVF test in the MoCA application. (**A**) The application will read the PVF instructions “please tell me as many words as possible that begin with the letter “ก” /k/ in one minute” when staff press the speaker button. (**B**) Space for staff to take notes. (**C**) Red letters show the timer. PVF, phonemic verbal fluency; MoCA, Montreal cognitive assessment.

**Figure 3 sensors-22-05813-f003:**
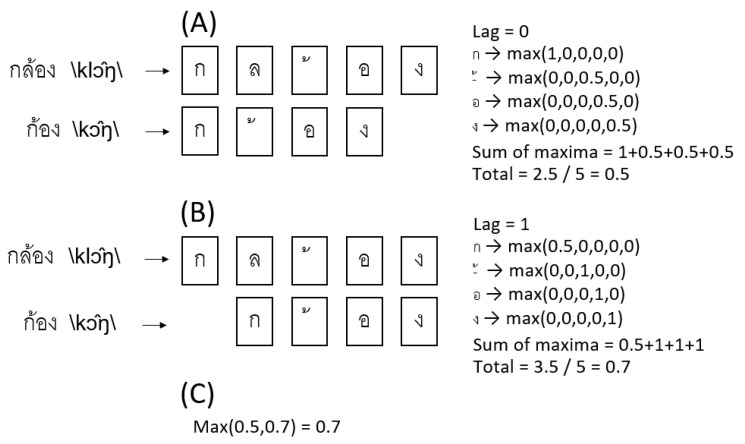
Illustration for orthographic similarity. (**A**) Words are placed at the same index to compare their letters. For calculating the maximum value, each letter in the shorter word is compared with the longest in every index. The quotient is 1/k, where k denotes the overlapped number of words index. The maximum quotients in each letter of the shorter word are summarized and divided by the longer word’s length. (**B**) The shorter word is shifted by one index; repeat the calculation of the maximum value. (**C**) Finding the maximum from the values obtained from every lag.

**Figure 4 sensors-22-05813-f004:**
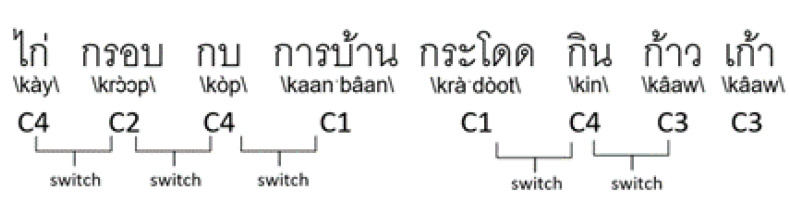
Illustration for switching.

**Figure 5 sensors-22-05813-f005:**
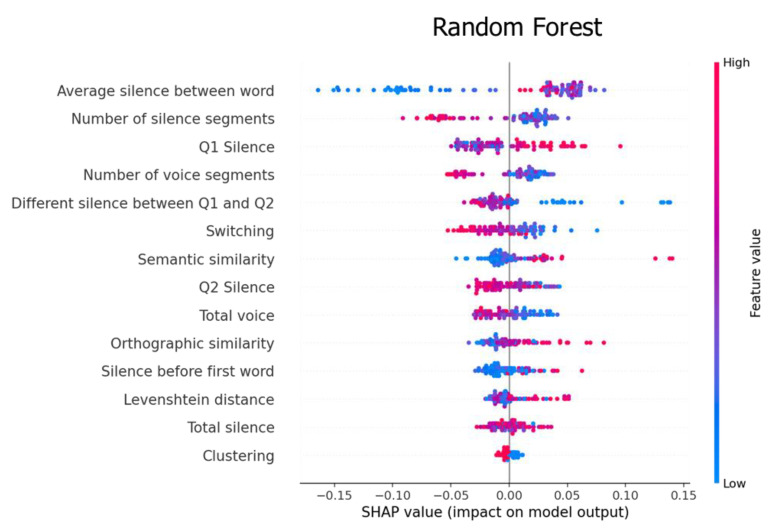
Feature importance explained by the SHAP value for the random forest classifier.

**Figure 6 sensors-22-05813-f006:**
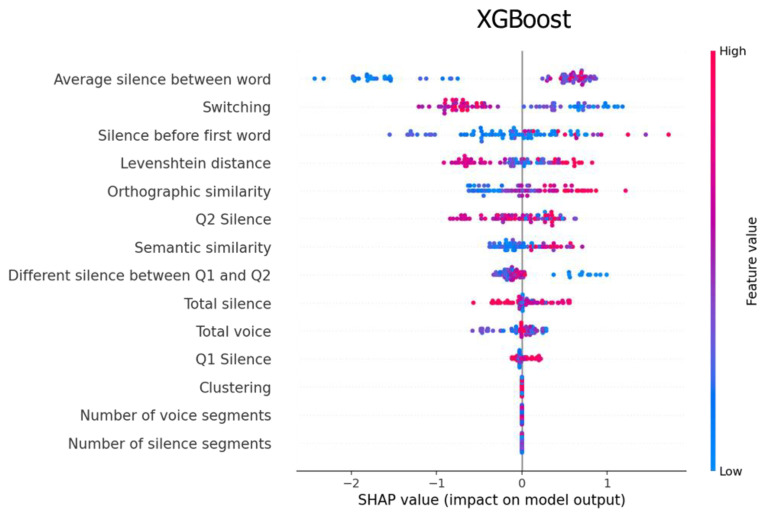
Feature importance explained by the SHAP value for the XGBoost classifier.

**Figure 7 sensors-22-05813-f007:**
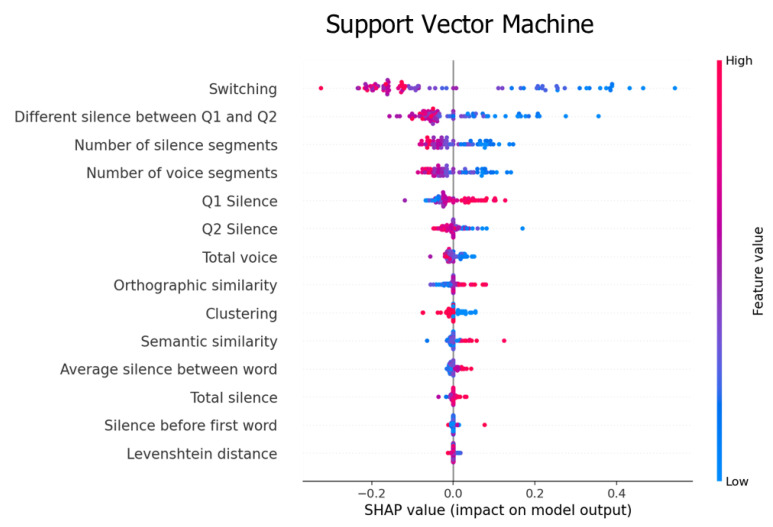
Feature importance explained by the SHAP value for the SVM classifier.

**Figure 8 sensors-22-05813-f008:**
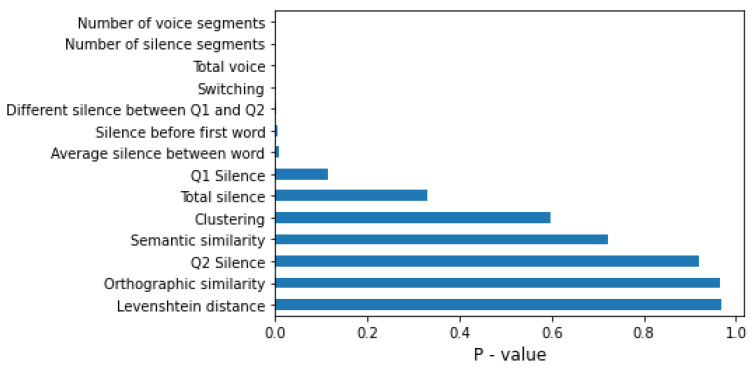
The *p*-values obtained from the chi-square test of the feature-selection process.

**Figure 9 sensors-22-05813-f009:**
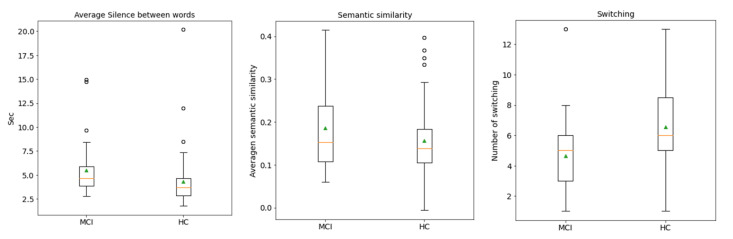
Distribution of the feature values between MCI and HC. Green triangles represent the data means. The orange lines show the medians of the data. White circles are the data outliers. MCI, mild cognitive impairment; HC, healthy control.

**Table 1 sensors-22-05813-t001:** Participant demographics.

	MCI (N = 41)	HC (N = 59)
Male	7	10
Female	34	49
Word count (mean)	3–15 (9.61)	2–24 (10.10)
MoCA Score (mean)	10–24 (21.59)	25–29 (27)

MCI, mild cognitive impairment; HC, healthy control; MoCA, Montreal cognitive assessment.

**Table 2 sensors-22-05813-t002:** Feature lists.

Feature	Description
Silence-based features	
Total silence	Total length of silence during the test.
Total voiced	Total length of voiced during the test.
Number of silence segments	Total number of silence segments.
Number of voice segments	Total number of voice segments.
Average silence between word	Total silence divided by the number of silence segments.
Q1 Silence	Total silence in the first 30 s of the audio file.
Q2 Silence	Total silence in the last 30 s of the audio file.
Silence before first word	Silence length before the participant speaks the first word.
Different silence between Q1 and Q2	Total silence in the first 30 s minus the last 30 s.
Similarity-based features	
Orthographic similarity	Average orthographic similarity value of all words.
Levenshtein distance	Average Levenshtein distance ratio of all words.
Semantic similarity	Average semantic similarity value of all words.
Cluster features	
Phonemic clustering	Group of words by phonemic categories.
Switching	Total number of the transition between clusters.

**Table 3 sensors-22-05813-t003:** The clusters in Thai.

Cluster	Characteristic	Example with IPA
1	Word started with “การ” /kaːn/ or “กะ” /kàʔ/ or “กระ” /kràʔ/	“การเรียน” /kaːn riːan/ “to learn”, “กระต่าย” /kràʔ tàːj/ “rabbit”, “กระโดด” /kràʔ dòːt/ “to jump”
2	Consonant blends	“กลวง” /kluːaŋ/ “hollow”, “กราบ” /kràːp/ “to pay respects”, “กวาด” /kwàːt/ “to sweep”
3	Homonym	“ก้าว” /kâːw/ “to step”, “เก้า” /kâːw/ “nine”
4	Word with only 1 syllable and others	“เกิด” /kə̀ət/ “born”, “แก่” /kɛ̀ɛ/ “old”, “เก็บ” /kèp/ “to store”

IPA, International Phonetic Alphabet.

**Table 4 sensors-22-05813-t004:** Classification results for the random forest classifier.

N	Acc.	F1-Score	Precision	Recall	Specificity	AUC
1	0.584 ± 0.16	0.565 ± 0.18	0.497 ± 0.24	0.535 ± 0.24	0.627 ± 0.22	0.636 ± 0.20
2	0.584 ± 0.16	0.565 ± 0.18	0.497 ± 0.24	0.535 ± 0.24	0.627 ± 0.22	0.636 ± 0.20
3	0.584 ± 0.18	0.561 ± 0.19	0.504 ± 0.26	0.530 ± 0.27	0.623 ± 0.24	0.629 ± 0.21
4	0.574 ± 0.19	0.556 ± 0.19	0.473 ± 0.21	0.510 ± 0.28	0.623 ± 0.19	0.649 ± 0.20
5	0.534 ± 0.20	0.501 ± 0.22	0.415 ± 0.29	0.375 ± 0.28	0.643 ± 0.19	0.660 ± 0.23
6	0.594 ± 0.20	0.563 ± 0.22	0.440 ± 0.26	0.450 ± 0.29	0.697 ± 0.18	0.653 ± 0.22
7	0.590 ± 0.18	0.558 ± 0.20	0.448 ± 0.26	0.500 ± 0.32	0.642 ± 0.17	0.646 ± 0.22
8	0.640 ± 0.23 *	0.616 ± 0.25 *	0.506 ± 0.30	0.575 ± 0.37 *	0.683 ± 0.17	0.667 ± 0.23
9	0.610 ± 0.20	0.579 ± 0.23	0.452 ± 0.28	0.550 ± 0.38	0.647 ± 0.15	0.650 ± 0.19
10	0.580 ± 0.19	0.552 ± 0.21	0.450 ± 0.28	0.455 ± 0.30	0.663 ± 0.15	0.671 ± 0.21
11	0.620 ± 0.21	0.600 ± 0.23	0.512 ± 0.30 *	0.530 ± 0.33	0.683 ± 0.17	0.683 ± 0.24 *
12	0.570 ± 0.18	0.545 ± 0.19	0.457 ± 0.24	0.455 ± 0.27	0.647 ± 0.15	0.642 ± 0.23
13	0.600 ± 0.17	0.565 ± 0.19	0.482 ± 0.27	0.430 ± 0.26	0.717 ± 0.15 *	0.642 ± 0.25
14	0.580 ± 0.19	0.542 ± 0.22	0.435 ± 0.32	0.430 ± 0.32	0.683 ± 0.17	0.617 ± 0.22

* The maximum value of each feature set; AUC, area under the receiver operating characteristic curve; Acc., accuracy; N, number of selected features, which has highest *p*-value by chi-square test.

**Table 5 sensors-22-05813-t005:** Classification results for the support vector machine classifier.

N	Acc.	F1-Score	Precision	Recall	Specificity	AUC
1	0.570 ± 0.15	0.557 ± 0.15	0.494 ± 0.14	0.610 ± 0.23*	0.543 ± 0.21	0.665 ± 0.23
2	0.570 ± 0.15	0.557 ± 0.15	0.494 ± 0.14	0.610 ± 0.23*	0.543 ± 0.21	0.669 ± 0.23
3	0.580 ± 0.17	0.563 ± 0.17	0.490 ± 0.17	0.540 ± 0.27	0.613 ± 0.18	0.672 ± 0.25
4	0.610 ± 0.19	0.588 ± 0.20	0.515 ± 0.25	0.505 ± 0.28	0.683 ± 0.17	0.680 ± 0.23
5	0.610 ± 0.21	0.576 ± 0.22	0.523 ± 0.29	0.430 ± 0.28	0.733 ± 0.20	0.717 ± 0.21
6	0.650 ± 0.22 *	0.626 ± 0.24 *	0.567 ± 0.28	0.525 ± 0.31	0.733 ± 0.20	0.721 ± 0.21
7	0.650 ± 0.21 *	0.624 ± 0.22	0.583 ± 0.28 *	0.505 ± 0.28	0.750 ± 0.20	0.729 ± 0.20
8	0.590 ± 0.18	0.551 ± 0.18	0.539 ± 0.29	0.365 ± 0.20	0.750 ± 0.20	0.725 ± 0.21
9	0.530 ± 0.11	0.362 ± 0.09	0.200 ± 0.40	0.025 ± 0.08	0.883 ± 0.17	0.733 ± 0.20 *
10	0.540 ± 0.11	0.366 ± 0.09	0.250 ± 0.43	0.025 ± 0.07	0.900 ± 0.17	0.733 ± 0.20
11	0.550 ± 0.07	0.356 ± 0.03	0.000 ± 0.00	0.000 ± 0.00	0.933 ± 0.11	0.733 ± 0.20
12	0.560 ± 0.07	0.358 ± 0.03	0.000 ± 0.00	0.000 ± 0.00	0.950 ± 0.11 *	0.725 ± 0.21
13	0.560 ± 0.07	0.358 ± 0.03	0.000 ± 0.00	0.000 ± 0.00	0.950 ± 0.11 *	0.725 ± 0.21
14	0.560 ± 0.07	0.358 ± 0.03	0.000 ± 0.00	0.000 ± 0.00	0.950 ± 0.11 *	0.725 ± 0.21

* The maximum value of each feature set; AUC, area under the receiver operating characteristic curve; Acc., accuracy; N, number of selected features, which has highest *p*-value by chi-square test.

**Table 6 sensors-22-05813-t006:** Classification results for the XGBoost classifier.

N	Acc.	F1-Score	Precision	Recall	Specificity	AUC
1	0.620 ± 0.15	0.594 ± 0.17	0.521 ± 0.24	0.605 ± 0.28 *	0.633 ± 0.22	0.640 ± 0.21
2	0.620 ± 0.15	0.594 ± 0.17	0.521 ± 0.24	0.605 ± 0.28 *	0.633 ± 0.22	0.640 ± 0.21
3	0.590 ± 0.17	0.558 ± 0.19	0.475 ± 0.24	0.480 ± 0.27	0.663 ± 0.19	0.626 ± 0.17
4	0.560 ± 0.14	0.515 ± 0.15	0.438 ± 0.21	0.390 ± 0.23	0.680 ± 0.20	0.659 ± 0.13
5	0.550 ± 0.17	0.497 ± 0.20	0.343 ± 0.28	0.400 ± 0.34	0.647 ± 0.19	0.550 ± 0.23
6	0.570 ± 0.17	0.540 ± 0.19	0.447 ± 0.24	0.480 ± 0.27	0.630 ± 0.21	0.638 ± 0.19
7	0.560 ± 0.17	0.526 ± 0.19	0.433 ± 0.22	0.450 ± 0.29	0.630 ± 0.19	0.617 ± 0.17
8	0.590 ± 0.20	0.564 ± 0.21	0.489 ± 0.23	0.500 ± 0.30	0.650 ± 0.22	0.621 ± 0.21
9	0.570 ± 0.13	0.536 ± 0.16	0.420 ± 0.22	0.455 ± 0.28	0.647 ± 0.13	0.638 ± 0.18
10	0.580 ± 0.14	0.550 ± 0.16	0.437 ± 0.22	0.480 ± 0.27	0.647 ± 0.13	0.642 ± 0.17
11	0.630 ± 0.11 *	0.603 ± 0.13 *	0.522 ± 0.16 *	0.530 ± 0.26	0.697 ± 0.09	0.642 ± 0.23
12	0.630 ± 0.18 *	0.585 ± 0.22	0.522 ± 0.35 *	0.455 ± 0.34	0.747 ± 0.15 *	0.650 ± 0.25
13	0.620 ± 0.17	0.592 ± 0.17	0.512 ± 0.24	0.505 ± 0.25	0.697 ± 0.14	0.671 ± 0.18 *
14	0.630 ± 0.13 *	0.601 ± 0.16	0.513 ± 0.23	0.505 ± 0.25	0.713 ± 0.10	0.629 ± 0.23

* The maximum value of each feature set; AUC, area under the receiver operating characteristic curve; Acc., accuracy; N, number of selected features, which has highest *p*-value by chi-square test.

## Data Availability

The data that support the findings of this study are available on request from the author, Hemrungrojn, S. The data are not publicly available due to ethic restrictions that their containing information that could compromise the privacy of research participants.

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
