# Peer review of "Selecting the Most Important Features for Predicting Mild Cognitive Impairment from Thai Verbal Fluency Assessments"

_sensors, 2022, doi:10.3390/s22155813_

Round 1

Reviewer 1 Report

It is considered that It is important to make the Thai version of PVF.

However, unfortunately, this study is not well-formed, particularly, contains no discussion. Furthermore, the purpose of the study needs to be presented more clearly.

In detail, in the introduction, it is explained that Thailand entered an aging society in 2001. However, the elderly population is rapidly increasing, so it would be good to add Thailand’s current situation.

This study used 2022-2022 data, and it would be good to clearly present whether the basis for the 20% of the MCI is 10-year data or current data.

Please provide the references at the beginning of the introduction section and sufficient description of the discussion in the discussion section. 

Reviewer 2 Report

This was a well written and well thought out paper that examined PVF for the Thai language with implications for detecting mild cognitive impairment.

Some minor comments:

1. Your paragraph ending line 78 and the aims paragraph, felt a little disconnected. Maybe a link that helps the reader understand how you plan to overcome these issues will help.  

2. Your results tables 4,5,6, i i believe N1 to 14 are features? This isnt very clear- maybe add what they are in the  legend?

3. Can Figure 5 be improved for resolution?

4. Disucssion is just a repeat of results. Please expand this to include references and links to other studies (like related words). So readers can understand better the implications for these findings. What other implications are there for the finsings? ie clinical?

5. Same with limitations- please expand rather than list- so that readers can understand what these limitations mean.

6. Same with Conclusion- what does this mean and what can it mean for future work? 

Reviewer 3 Report

The paper proposes some mechanisms for analysis phonetic verbal fluency (PVF) through analytical process feature extraction.  Although the problem is indeed important, there are several issued that must be attended which are:

1.        The contribution of the paper is not clear.  It is necessary clarify it to understand the significance o the paper.

2.       According to the system shown in Figure 1, the input data are audio signals and transcribed file, however there is not any explanation about how the audio signals are segmented.  It would be useful to include a detailed explanation of the system shown in Figure 1.

3.       The feature extraction stage requires more detail.

4.       In several parts of the paper it is mentioned the switching, however it is not mentioned why the switching is important.

5.       In equation (1) it is mentioned that Oi is the observed vales.  Please explain if this observed value is a segmented audio signal or a transcribed data file or both.

6.       It is necessary to include the mathematical expressions of the variables in Tables 4-6. Acc, F1-Score, Precision, etc.

7.       Please comment the results included in Figure 5.  Without an explanation it is not possible to understand the importance of these results.

8.       Please explain how the letters of the input words are segmented in the clustering algorithm shown in Fig. 8.

9.       Please improve the conclusions of the paper, indication how the proposed method compares with other similar schemes.        

Round 2

Reviewer 1 Report

Please conform the basic format of the paper as a whole.

In the abstract, it would be recommended to briefly describe the background and present the research purpose more clearly.

In the introduction, I hope that the purpose of the study should be presented more clearly and that the research results and national methods should be excluded.

Thank you so much.

Author Response

Dear Editors,

Thank you for your valuable comments. I have revised my manuscript as you suggested.
I hope this improvement will be sufficient.

Best Regards,
Suppat Metarugcheep

Reviewer 3 Report

The authors attened the revierwer comments, then I consider taht the paper can be accepted in its actual form

Author Response

Dear Editors,

Thank you for your valuable comments. I have formatted my manuscript into basic format.
I hope this improvement will be sufficient.

Best Regards,
Suppat Metarugcheep